# Learning Conditional Invariances through Non-Commutativity

**Abhra Chaudhuri**[1,2,3]
[1] University of Exeter

**Serban Georgescu** [2]
[2] Fujitsu Research of Europe

**Anjan Dutta**[3] *
[3] University of Surrey

## Abstract

Invariance learning algorithms that conditionally filter out domain-specific random variables as distractors, do so based only on the data semantics, and not the target domain under evaluation. We show that a provably optimal and sample-efficient way of learning conditional invariances is by relaxing the invariance criterion to be non-commutatively directed towards the target domain. Under domain asymmetry, *i.e.*, when the target domain contains semantically relevant information absent in the source, the risk of the encoder $\varphi^*$ that is optimal on average across domains is strictly lower-bounded by the risk of the target-specific optimal encoder $\Phi_\tau^*$. We prove that non-commutativity steers the optimization towards $\Phi_\tau^*$ instead of $\varphi^*$, bringing the $\mathcal{H}$-divergence between domains down to zero, leading to a stricter bound on the target risk. Both our theory and experiments demonstrate that non-commutative invariance (NCI) can leverage source domain samples to meet the sample complexity needs of learning $\Phi_\tau^*$, surpassing SOTA invariance learning algorithms for domain adaptation, at times by over 2%, approaching the performance of an oracle. Implementation is available at https://github.com/abhrac/nci.

## 1 Introduction

The idea of learning robust representations by identifying invariants across domains/views has found applications in several areas of machine learning including domain adaptation (Ganin & Lempitsky, 2015; Arjovsky et al., 2019), self-supervised learning (Federici et al., 2020; Chavhan et al., 2023), and causal inference (Pogodin et al., 2023; Peters et al., 2016) to name a few. The traditional way of learning such representations is based on the assumption that domain shifts are semantically irrelevant transformations $\mathcal{T}$ applied to a shared, underlying concept $\varphi(\mathbf{x})$ (Nguyen et al., 2021; Qiao et al., 2020; Volpi et al., 2018). The process of invariance learning would thus be formulated as identifying the marginal distribution $P(\varphi(\mathbf{x}^d))$ that matches across all training domains $d$ (Ganin & Lempitsky, 2015; Rosenfeld et al., 2021).

A number of works, however, also identified that estimating the above marginal is not sufficient, as different classes have different invariance requirements. Instead, a conditional form of invariance that matches the distribution of $P(\varphi(\mathbf{x}^d)|\mathbf{y})$ for all labels $\mathbf{y}$ across domains is a more realistic solution, an idea which came to be known as conditional invariance (Li et al., 2018; Long et al., 2018; Samarin et al., 2021). This was further studied by Stojanov et al. (2021), who narrowed down the reason for the requirement of conditional invariance as follows –

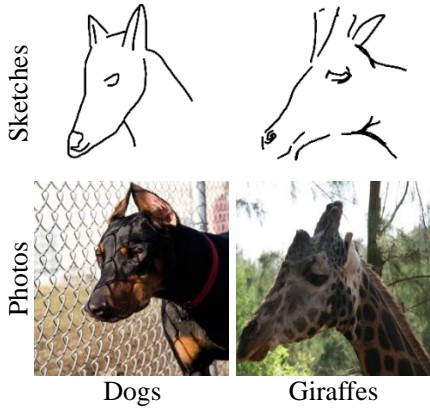

Sketches

Photos

Dogs          Giraffes

Figure 1: Dogs and giraffes may not be fully distinguishable in the free-hand sketch domain due to shared geometric structures. However, they are clearly separable in the photo domain based on color and texture. Thus, mapping photos and sketches to the same invariant representation loses out such critical domain-specific, but useful information when inference has to be performed in the photo domain at test time.

---

*Abhra Chaudhuri (ac1151@exeter.ac.uk) is the corresponding author.

tasks not only depend on the shared information across domains, but also domain specific attributes, that can be retained only when the representation learning is conditioned as $P(\varphi(x)|\mathbf{y})$, as opposed to the traditional marginal-only learning of $P(\varphi(\mathbf{x}))$. This reason was also echoed in the work of Eastwood et al. (2022), based on the fact that encoders that perform better on average across domains do not do well on specific domains (Eastwood et al., 2022), due to the No Free Lunch Theorem (Wolpert & Macready, 1997). Despite its promise, the idea of retaining domain-specific features for identifying meaningful invariants has remained relatively unexplored since Stojanov et al. (2021), with some applications to tasks like cross-modal retrieval (Chaudhuri et al., 2022).

The necessity of conditional invariance is observable when the target domain contains more semantically relevant information than the source domain, a condition we refer to as Asymmetry (Assumption 1). This phenomenon is depicted with an example in Figure 1 from the PACS dataset (Li et al., 2017). In the free hand sketch domain, dogs and giraffes may look very similar in scenarios where the full body is not drawn. But in real photos, they can be easily distinguished based on colors and textures. Mapping photos and sketches to an unconditionally invariant representation space gets rid of such useful domain-specific information, retaining only the shared aspects like object geometry. This significantly harms prediction performance when the target domain is that of photos, which can potentially provide asymmetrically greater semantic information relative to sketches.

We provide a novel learning-theoretic view based on commutativity that establishes why unconditionally invariant representations exhibit such limitations. We do so by framing the process of invariance learning as discovering operators, the choice of which determines whether the set of source and target domains form a commutative group. As a consequence of our formulation, we observe that a significantly simpler and efficient approach to achieving conditional invariance is by relaxing the invariance criterion (in approaches that learn the shared marginal $P(\varphi(\mathbf{x}))$ such as Domain Adversarial Training (Ganin & Lempitsky, 2015)) to be non-commutatively directed towards the target domain. One of the primary bases of our results is that, for a hypothesis class $\mathcal{H}_\varphi$, the risk of the encoder $\varphi^* \in \mathcal{H}_\varphi$ that is optimal across domains is lower bounded by the risk of the target-specific encoder $\Phi_\tau^*$ that is optimal for the target, but not necessarily for other domains. When training samples from the source domain are semantically complementary to the ones in the target domain (for example, photos and sketches of different objects, rather than that of the same object), we find that non-commutative invariance (NCI) can leverage source domain samples to meet the sample complexity needs of learning $\Phi_\tau^*$. NCI provides a stricter upper bound on target risk relative to existing unconditional (Ganin et al., 2016) and conditional (Gulrajani & Lopez-Paz, 2021; Stojanov et al., 2021) paradigms through a lower value of $\mathcal{H}$-divergence (Ben-David et al., 2006; 2010; Kifer et al., 2004; Ganin et al., 2016).

Intuitively, NCI produces representations such that samples from the source domain get mapped to the representation space of the target domain, while those from the target domain continue to live in the representation space of the target domain. This allows the retention of task-relevant target domain features, while still being able to leverage the semantic augmentations from the training source domains. As inference needs to be performed in the target domain at test-time, this is a desirable property. We back our theory with a number of empirical results which establish how NCI can leverage semantic complementarity across domains to achieve SOTA performance in domain adaptation, approaching an oracle that has access to both source and target domains at test time.

In summary, we (1) show that conditional invariance can be achieved by simply relaxing the invariance operator to be non-commutative; (2) derive stricter and efficient generalization and sample complexity bounds for learning the optimal target encoder; (3) conduct experiments demonstrating that with the right direction of invariance, non-commutativity can leverage complementary semantic information across domains to achieve SOTA performance in domain adaptation.

## 2 RELATED WORK

**Invariance Learning and Domain Adaptation:** State-of-the-art in domain adaptation can be primarily classified into approaches that (1) learn invariant features that are robust to domain shifts (Ganin & Lempitsky, 2015; Ganin et al., 2016); (2) represent domains as learnable transformations of the shared, underlying semantics (Nguyen et al., 2021; Qiao et al., 2020; Volpi et al., 2018); and (3) smoothen the loss landscape (Rame et al., 2022; Rangwani et al., 2022; Wortsman et al., 2022), or by penalizing localized features (Wang et al., 2019). Our work primarily belongs to (1),

which is what we will focus on in this section. The idea of learning invariant features for domain adaptation (DA) was first proposed in Ganin & Lempitsky (2015); Ganin et al. (2016) via gradient reversal. A causal take on invariance learning was proposed by Peters et al. (2016) with the view that the underlying causal factors remain the same under domain shift, which also constitutes the foundation for probable domain generalization (Eastwood et al., 2022). Arjovsky et al. (2019) concretely instantiated this idea through Invariant Risk Minimization (IRM), which was followed by several subsequent extensions (Wang et al., 2022; Lu et al., 2022). However, Rosenfeld et al. (2021) showed that when the training and test distributions sufficiently diverge, nothing is fundamentally better than vanilla Empirical Risk Minimization (ERM), a claim that was experimentally supported by Gulrajani & Lopez-Paz (2021). Other promising approaches to invariance learning involve imposing various regularizers based on the information bottleneck principle (Tishby et al., 2000; Federici et al., 2020; Ahuja et al., 2021). However, the above approaches discard all domain-specific information, retaining only the shared features. It was consequently discovered that learning a task dependent, conditional form of invariance instead of marginal invariance is much more beneficial, as certain domain-specific features may actually be helpful for downstream tasks (Pogodin et al., 2023; Stojanov et al., 2021; Chavhan et al., 2023; Quinzan et al., 2022). Our work further explores this idea through the lens of commutativity of domains under the invariance operator, and presents a simple and efficient approach for conditional invariance learning. We also show that our approach can leverage source domain samples to meet the sample complexity needs of generalization to the target domain (Theorem 2). So, we provide an extended literature review on approaches that endeavour to achieve the same in Appendix A.1.

## 3 Non-Commutative Invariance

### 3.1 Preliminaries

**Overview:** We provide the complete set of general notations in Appendix A.2. We start by introducing the notion of commutativity, which is an abstract generalization of a large family of marginal invariance learning paradigms like Domain Adversarial Neural Networks (DANNs) (Ganin & Lempitsky, 2015), Invariant Risk Minimization (Arjovsky et al., 2019; Ahuja et al., 2021), and several of their derivatives under DomainBed (Gulrajani & Lopez-Paz, 2021). The idea of commutativity allows us to identify useful, domain-specific features that may be discarded by such marginal invariance learning algorithms. As a remedy, we define the dual notion of Non-Commutative Invariance (NCI), which we find to has stronger theoretical properties in terms of (1) a stricter generalization bound on the target risk under an asymmetry condition; and (2) the ability to meet the sample complexity for learning the optimal target domain encoder $\Phi_\tau^*$ with samples from the source domain. Associated preliminaries and proofs are provided in Appendix A.3 and Appendix A.4 respectively.

**Definition 1** (Asymmetry). Domains $\mathcal{D}_s$ and $\mathcal{D}_\tau$ are said to be asymmetric *iff*:

$$\mathbf{x}_s = \mathcal{T}_s(\mathbf{x}) \ \text{and} \ \mathbf{x}_\tau = \mathcal{T}_\tau(\mathbf{x}) \ \not\Rightarrow \ I(\Phi_s^*(\mathbf{x}_s); \mathbf{y}) = I(\Phi_\tau^*(\mathbf{x}_\tau); \mathbf{y})$$

where $\mathbf{x}_s \sim \mathcal{D}_s$, $\mathbf{x}_\tau \sim \mathcal{D}_\tau$, and $\mathcal{T}_s$ and $\mathcal{T}_\tau$ are transformations applied on the underlying concept $\mathbf{x} \sim X$ to derive such domain-specific instantiations.

Even though $\mathbf{x}_s$ and $\mathbf{x}_\tau$ may be derived as transformations of the same underlying concept $\mathbf{x}$ (Nguyen et al., 2021; Qiao et al., 2020; Volpi et al., 2018), it is not necessary that the two domains would contain the same amount of label information $\mathbf{y}$. This is because different transformations $\mathcal{T}_s(\mathbf{x})$ and $\mathcal{T}_\tau(\mathbf{x})$ may differ in the degree to which they preserve the label information in $\mathbf{x}$. The optimal encoder $\Phi_{s/\tau}^*$ of the respective domain $s/\tau$ would be able reflect this asymmetry by capturing different amounts of label information. As a result, performing a downstream task with the optimal encoding would result in different accuracies across the two domains. In other words, the domain-specific optimal encoding $\Phi^*(\mathbf{x})_{s/\tau}$ is not invariant to the domain information.

**Assumption 1.** For the remainder of our analysis, we assume that the target domain contains more semantically relevant information than the source domain, *i.e.*,

$$I(\Phi_\tau^*(\mathbf{x}_\tau); \mathbf{y}) > I(\Phi_s^*(\mathbf{x}_s); \mathbf{y}),$$

thus warranting the preservation of target domain specific information. This assumption quantifies the example illustration in Figure 1.

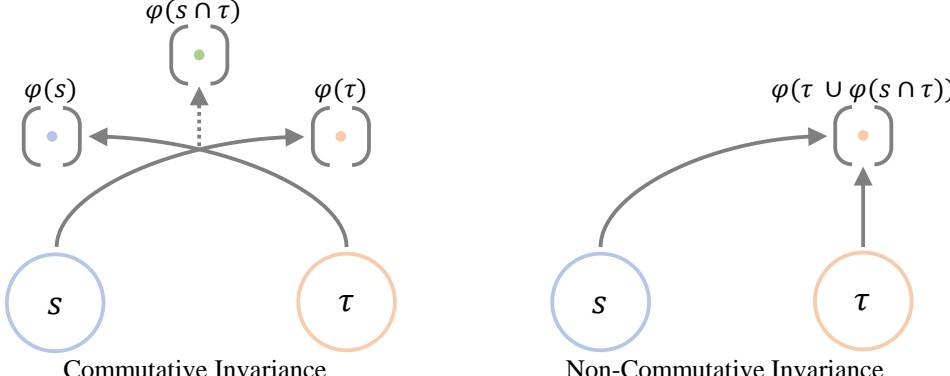

Figure 2: Abstract comparison of commutative and non-commutative invariance (NCI) learning. Commutative invariance aims to capture components that are shared across both the source $\mathcal{D}_s$ and target $\mathcal{D}_\tau$ domains (simplified in the diagram as $s$ and $\tau$ respectively) by mapping samples from one domain to the semantic space $\varphi(\cdot)$ of the other. NCI, on the other hand, captures components that the source domain shares with the target domain by mapping source samples to the target's representation space, but retains all the components in the semantic space of the target domain unchanged. NCI is thus invariant to the domain-specific, semantically relevant components of the source domain, but not the target domain.

**Definition 2** (Commutative and Non-Commutative Invariance). Let $\mathbf{x}_s \sim \mathcal{D}_s$ and $\mathbf{x}_\tau \sim \mathcal{D}_\tau$ be domain-specific samples for an instance $\mathbf{x}$. We call the operator $\otimes$ commutatively invariant *iff*:

$$\mathbf{x}_s \otimes \mathbf{x}_\tau = \mathbf{x}_\tau \otimes \mathbf{x}_s = \varphi^*(\mathbf{x}),$$

*i.e.*, $\otimes$ produces the same output as $\varphi^*(\cdot)$ that is optimal when the downstream task is performed on the the joint distribution $(\mathcal{D}_s, \mathcal{D}_\tau)$ (but not necessarily optimal in the individual domains, due to the No Free Lunch Theorem Wolpert & Macready (1997)), and hence is independent of the order of the operands. Algorithms that try to recover the marginal distribution $P(X)$, eliminating all domain-specific information across both $\mathcal{D}_s$ and $\mathcal{D}_\tau$ in the process (Ganin & Lempitsky, 2015; Arjovsky et al., 2019; Ahuja et al., 2021) fall in this category. Essentially, they induce a commutative semigroup on the set of domains under $\otimes$ (Theorem 4). Now, the operator $\otimes$ is right-invariant *iff*:

$$\mathbf{x}_s \otimes \mathbf{x}_\tau = \varphi^*_s(\mathbf{x}) \quad \text{and} \quad \mathbf{x}_\tau \otimes \mathbf{x}_s = \varphi^*_\tau(\mathbf{x})$$

$\otimes$ is right-invariant as it always returns an encoding of $\mathbf{x}$ that is optimal in the domain of the left operand, no matter what domain the right operand comes from. A notion of left-invariance can be symmetrically defined. We call such operators that are invariant to the domain of one operand, but is sensitive to the domain of the other, non-commutatively invariant (NCI). Figure 2 graphically depicts an abstract comparison between commutative invariance and NCI. An associated notion is that of the discrepancy between the domains that $\otimes$ aims to reduce, which we discuss below.

$\mathcal{H}_\eta$**-divergence and generalization bound on the target risk:** The $\mathcal{H}_\eta$-divergence between $\mathcal{D}_s$ and $\mathcal{D}_\tau$ is a measure of the capacity of the discriminator hypothesis class $\mathcal{H}_\eta$ to distinguish between the source and the target sample representations. It has found use in a number of formal treatments of domain adaptation (Ben-David et al., 2006; 2010; Kifer et al., 2004; Ganin et al., 2016). It is used to quantify the discrepancy between domains, often in the representation space of an encoder $\varphi(\cdot)$, and is proportional to the difference in domain-specific information between $\mathcal{D}_s$ and $\mathcal{D}_\tau$. Primarily, it was used by Ben-David et al. (2006) to derive an upper bound for the target risk $R_{\mathcal{D}_\tau}(\cdot)$ (test-time error in the target domain) of an encoder $\varphi(\cdot)$ that is trained with labelled source and unlabelled target samples. The complete definition of $\mathcal{H}$-divergence, along with its empirical estimate, and the associated generalization bound on the target risk are provided in Definition 5 and Theorem 3 respectively. Below, we show that the bound in Theorem 3 can achieve a stricter form under a non-commutatively invariant encoding of the domains.

## 3.2 Target Risk under Non-Commutativity

**Result 1.** If the VC (Vapnik–Chervonenkis) dimension (Vapnik & Chervonenkis, 1971) of $\mathcal{H}_\varphi$, $VC(\mathcal{H}_\varphi) \geq N$, the total number of training samples, then the target risk of the optimal NCI encoder

$R_{\mathcal{D}_\tau}(\varphi_\tau^*)$ is related to the target risk of the commutatively invariant encoder $R_{\mathcal{D}_\tau}(\varphi^*)$ as:

$$R_{\mathcal{D}_\tau}(\varphi_\tau^*) \leq R_{\mathcal{D}_\tau}(\varphi^*) - \hat{d}_{\mathcal{H}_\eta}(S, T, \varphi^*) < R_{\mathcal{D}_\tau}(\varphi^*),$$

where $\hat{d}_{\mathcal{H}_\eta}$ is the empirical $\mathcal{H}_\eta$-divergence between source and target samples $S$ and $T$ respectively.

The proof of Result 1 is based on the observation that $\hat{d}_{\mathcal{H}}(S, T, \varphi)$ does not reach 0 if $\varphi(\cdot)$ is trained with commutative invariance. Commutative invariance maps data from the source to the target domain and vice versa thereby solving a swapped prediction problem. That way, although the "min" part of Equation (5), which quantifies the generalization error (Definition 6), takes a value of 0, the overall $\mathcal{H}_\eta$-divergence does not become 0. To bring $\hat{d}_{\mathcal{H}_\eta}(S, T, \varphi)$ down to 0, the objective in the "min" part has to evaluate to 1. One way to achieve that is by finding a $\varphi(\cdot)$ that maps all the data to the source domain. Although Result 1 provides a stricter generalization bound on the target risk, it does not establish the practicality of non-commutativity. Below, we show that it is possible to give further guarantees for the optimality of non-commutativity when test-time evaluation is performed in the target domain. We establish that training with NCI results on samples from the source domain being treated as augmentations in the target domain. As a consequence, as the number of source samples increase, the NCI encoder $\varphi_\tau^*(\cdot)$ approaches the optimal encoder for the target domain, which we denote by $\Phi_\tau^*$.

## 3.3 SAMPLE COMPLEXITY OF NCI AND ITS OPTIMALITY

We start this section by exploring the relationship between the target risks of the global encoder $\varphi^*$ that is optimal on average in $\mathcal{H}_\varphi$, and the optimal encoder for the target domain $\Phi_\tau^*$. We show that $R(\varphi^*)$ is no less than the risk of the optimal encoder for the target domain $R(\Phi_\tau^*)$. We finally demonstrate how non-commutativity surpasses this bound and achieves the optimal risk for the target domain $R(\Phi_\tau^*)$, by meeting its sample-complexity needs with source domain samples.

**Theorem 1.** *Let the $\mathcal{H}_\eta$-divergence between the source ($s$) and the target ($\tau$) domains be $\Delta$, i.e., $s = \tau + \Delta$ (Appendix A.3.1). Under asymmetry (Assumption 1), i.e., $I(\Phi_s^*(\mathbf{x}_s); \mathbf{y}) < I(\Phi_\tau^*(\mathbf{x}_\tau); \mathbf{y})$, the risk of the optimal-on-average encoder $R(\varphi^*)$ for predicting $Y$ across all domains up to a distance $\Delta$ from the domain $\tau$ admits the following upper bound:*

$$R(\varphi^*) = \min_{\varphi^* \in \mathcal{H}_\varphi} \frac{1}{\Delta} \int_{\theta=\tau}^{\tau+\Delta} l\left(\varphi^*(\mathcal{D}_\theta), Y\right) dP(\theta, Y) \geq l\left(\Phi_\tau^*(\mathcal{D}_\tau), Y\right),$$

*where $l(\cdot, \cdot)$ calculates the encoding risk of a domain, and $\Phi_\tau^*(\cdot)$ is the optimal encoder for $\tau$.*

From Lemma 3, we know, the average risk is greater than the shared risk, *i.e.*, $R(\varphi^*) \geq \min_{\varphi^* \in \mathcal{H}_\varphi} l(\varphi^*(\mathcal{D}_s \cap \mathcal{D}_\tau), Y)$. Now, the shared risk has to be greater than or equal to the maximum of the source risk and the target risk. If the target risk is greater than the source risk, then the shared risk is at least large as the target risk. If the source risk is greater than the target risk, then the shared risk is strictly greater than the target risk. In other words, the optimal-on-average predictor $\varphi^*$ across all domains up to a distance $\Delta$ from the target domain $\tau$ cannot do any better than the optimal predictor of the target domain $\varphi_\tau^*$. This means that learning the globally optimal predictor is a sub-optimal solution when the downstream task has to be performed in a single target domain $\tau$. The bound for $R(\varphi^*)$ in Theorem 1 becomes strictly greater when the transformation $\mathcal{T}$ from which the target domain $\tau$ was obtained preserves the causal factors for the label $\mathbf{y}$ of $\mathbf{x}$, and there exists some domain in the distance $d \leq \Delta$ from $\tau$ where the causal factors are not preserved, *i.e.*:

$$\mathbf{x}_\tau = \mathcal{T}_\tau(\mathbf{x}) \mid I(\mathbf{x}, \mathbf{y}) = I(\mathbf{x}_\tau, \mathbf{y}) \text{ and } \exists \mathcal{T}_s \in [\tau, \tau + d] \mid \mathbf{x}_s = \mathcal{T}_s(\mathbf{x}), I(\mathbf{x}, \mathbf{y}) > I(\mathbf{x}_s, \mathbf{y}), \quad (1)$$

where $\mathbf{x}$ lies in the ground-truth concept basis $\mathbb{C}$ (Lemma 1). Hence, looking for the optimal predictor specifically for the target domain is always guaranteed to give at least as good a solution for domain adaptation, if not better. The below theorem establishes that NCI solves this very problem by leveraging samples from the source domain to meet the sample complexity needs of learning the optimal target encoder $\Phi_\tau^*$, and exhibits the same error bound on $\mathcal{D}_\tau$ as that of $\Phi_\tau^*$.

**Theorem 2.** *Consider a function class $\mathcal{T}$ of transformations that defines target domains such that $\tau \sim \mathcal{T}, \mathbf{x}_\tau = \tau(\mathbf{x}), \mathbf{x}_\tau \sim \mathcal{D}_\tau$. Let $\Phi_\tau^*(\cdot)$ be the optimal encoder for the target domain with*

*sample complexity for incurring an error of $\epsilon$ with probability $\delta$. From Haussler's theorem, the error incurred by the optimal target domain encoder $\Phi_\tau^*$ with probability $\delta$ and $M$ target domain samples is given by:*

$$\epsilon \geq \frac{\ln|\mathcal{T}| + \ln\frac{1}{\delta}}{M},$$

*With $m_\tau$ target domain samples and $m_s$ source domains samples, such that $M \leq m_s + m_\tau, M > m_\tau$, the optimal NCI encoder $\varphi_\tau^*(\cdot)$ achieves the same error bound $\epsilon$ with probability $\delta$.*

Note that the constraints on the sample size in Theorem 2 requires the number of source samples to be $> 0$. When $m_\tau$ samples from the target alone cannot attain the sample complexity $M$ for achieving the optimal error bound $\epsilon$ of $\Phi_\tau^*$, NCI can leverage $m_s$ novel samples from the source domain to achieve the same error bound $\epsilon$ as that of $\Phi_\tau^*$. This suggests that NCI treats the additional samples from the source domain as augmentations in the target, a property that is also reflected in the proof of Theorem 2. Note that the constraints on the sample size in Theorem 2 requires the number of both source and target samples to be $> 0$, and the total number of source and target samples must at least add up to achieve the sample complexity of $\Phi_\tau^*$. Thus, the source and target samples must each come from a unique support in order to qualify as distinct. We empirically validate this fact in Section 4 by varying degrees of complementary (and shared) information between the source and the target domains, and tracking its effects on classification accuracy.

With the above results, although we can guarantee the optimality of NCI, we cannot guarantee its uniqueness under the given semantic constraints. This is because there could be multiple ways to bring the source-target $\mathcal{H}_\eta$-divergence down to 0, which may be dependent on the constraints imposed by the downstream task. A related finding by Volpi et al. (2018) shows that domain-shifted augmentations penalize deviations from the parameter vector corresponding to that of the true label instead of regularizing towards zero unlike classic regularizers. We conjecture that this property also applies to NCI, and can be an alternative approach to proving Theorem 2, and perhaps, understanding the uniqueness of the solution.

## 3.4 TRAINING WITH NCI

Below we provide an example of converting a commutative invariance objective into its non-commutative form. We choose the Domain Adversarial Neural Network (Ganin & Lempitsky, 2015) as the candidate for this example because of its simplicity and ubiquity across the conditional invariance learning literature (Long et al., 2018; Li et al., 2018; Stojanov et al., 2021). The discriminator $\eta(\cdot)$ is trained to distinguish between the source and the target domain from the representation space of $\varphi_\tau$ by minimizing the following:

$$\mathcal{L}_\eta = \mathbf{y}\log\eta(\varphi_\tau(\mathbf{x})) + (1 - \mathbf{y})\log(1 - \eta(\varphi_\tau(\mathbf{x}))) \tag{2}$$

The encoder for the target domain $\varphi_\tau$ is trained to maximize the $\mathcal{L}_\eta$ through non-commutatively mapping the source samples to the target domain by minimizing the following objective:

$$\mathcal{L}_\varphi = (1 - \mathbf{y})\log(\eta(\varphi_\tau(\mathbf{x}))) \tag{3}$$

Note that the above objective is non-commutatively invariant to the source domain because the loss is only minimized for source domain samples with label 0, and not for the target domain samples with label 1. In other words, $\varphi(\cdot)$ learns representations such that the discriminator sees source domain samples as being from the target domain, but not the other way around. Optimizing the above set of objectives should thus lead to the convergence of $\varphi(\cdot)$ towards the optimal encoder for the target domain $\varphi_\tau^*(\cdot)$ according to Theorem 2.

## 4 EXPERIMENTS

**Datasets and Experimental Settings:** We perform experiments with NCI on three standard domain adaptation benchmarks, namely, PACS (Li et al., 2017), Office-Home (Venkateswara et al., 2017), and DomainNet (Peng et al., 2019). We evaluate NCI on the task of multi-source domain adaptation (Zhao et al., 2018) experiments with complementary semantics across domains. We model this setting by not sharing the full instance support across domains. Specifically for PACS and Office-Home, 70% of the sample supports are shared between both sources and targets, and each of the

| Method | PACS | | | | Office-Home | | | |
|---|---|---|---|---|---|---|---|---|
| | **Photo** | **Art** | **Cartoon** | **Sketch** | **Photo** | **Clipart** | **Product** | **Real** |
| ERM (Wiley 1998) | 81.33 | 77.60 | 88.76 | 78.30 | 49.15 | 45.66 | 54.32 | 60.60 |
| DANN (ICML'15) | 78.86 | 75.32 | 90.37 | 80.66 | 47.31 | 46.15 | 54.79 | 58.17 |
| CDANN (ECCV'18) | 78.95 | 76.72 | 90.02 | 80.00 | 47.64 | 46.02 | 54.50 | 59.45 |
| MDAN (NeurIPS'18) | 79.37 | 77.05 | 88.90 | 79.15 | 48.00 | 45.77 | 54.45 | 60.90 |
| MDD (ICML'19) | 79.55 | 77.62 | 87.61 | 79.48 | 47.99 | 45.31 | 54.37 | 59.16 |
| CICyc (GCPR'21) | 80.02 | 76.67 | 89.35 | 78.86 | 48.96 | 45.50 | 55.11 | 60.02 |
| IB-IRM (NeurIPS'21) | 77.01 | 75.11 | 90.78 | 80.95 | 45.40 | 46.91 | 55.25 | 57.96 |
| EQRM (NeurIPS'22) | 80.35 | 78.00 | 90.11 | 78.10 | 49.55 | 45.10 | 55.21 | 60.90 |
| CIRCE (ICLR'23) | 80.78 | 79.48 | 89.72 | 80.55 | 48.20 | 44.97 | 54.21 | 60.75 |
| SDAT+ELS (ICLR'23) | 81.25 | 80.21 | 89.32 | 80.27 | 48.50 | 45.43 | 55.39 | 60.97 |
| **NCI (Ours)** | **83.40** | **81.55** | **91.05** | **81.37** | **50.02** | **47.90** | **56.00** | **63.50** |

Table 1: Comparison with SOTA invariance learning approaches on PACS and Office-Home.

| Method | PACS | | | | Office-Home | | | |
|---|---|---|---|---|---|---|---|---|
| | **Photo** | **Art** | **Cartoon** | **Sketch** | **Photo** | **Clipart** | **Product** | **Real** |
| Fishr (ICML'22) | 81.19 | 80.06 | 89.05 | 79.68 | 48.25 | 45.15 | 54.39 | 59.92 |
| + NCI | **84.90** | **82.22** | **92.00** | **82.55** | **51.37** | **48.07** | **56.86** | **64.02** |
| SDAT (ICML'22) | 81.00 | 79.95 | 88.11 | 79.32 | 48.07 | 45.20 | 54.55 | 60.37 |
| + NCI | **83.80** | **82.06** | **91.92** | **82.07** | **51.44** | **48.05** | **56.95** | **64.22** |
| Model Soups (ICML'22) | 81.26 | 79.37 | 89.76 | 79.05 | 48.55 | 45.71 | 54.48 | 60.91 |
| + NCI | **85.09** | **82.56** | **92.77** | **82.62** | **51.95** | **48.49** | **57.20** | **64.50** |

Table 2: Accuracy gains from NCI on top of flat-minima based OoD generalization algorithms.

3 targets provide 10% unique data support instances. For DomainNet, 75% of the instances are shared across domains, and each of the 5 source domains provide 5% novel support instances. For comparing NCI against an oracle that has access to all the domains at test time, we perform a separate set of experiments on the multi-modal material segmentation task introduced by Liang et al. (2022) on their proposed dataset. For all our experiments, we use the NCI version of Domain Adversarial Neural Networks (DANN) (Ganin & Lempitsky, 2015), as described in Section 3.4, with all hyperparameters and experimental settings based on the DomainBed (Gulrajani & Lopez-Paz, 2021) version of DANN, as that provides a fair ground for comparison with existing arts. Through our experiments, we not only aim to verify the domain adaptation performance of NCI relative to SOTA, but also our theory (Theorem 2) on the ability of NCI to leverage samples of complementary semantics from source domains to learn the optimal target encoder, along with our Asymmetry assumption (Assumption 1). We also wish to understand how NCI compares with an oracle that has the liberty to optimally choose useful features from all domains at test time.

**Comparison with SOTA invariance learning approaches:** We report the performance of NCI compared to existing SOTA invariance learning algorithms in Table 1 (PACS and Office-Home) and Table 4 (DomainNet). We provide a discussion on the algorithms that we compare NCI with, as well as the results on DomainNet in Appendix A.5. The observations agree with our premise that invariance to all domain-specific information may or may not be helpful depending on the target domain. For instance, in Table 1, although DANN outperforms ERM in Cartoon and Sketch in PACS, it lags behind in Photo and Art by similar margins. This indicates that the domain-specific information it "discards" is somehow helpful for classification of Photos and Art paintings. This happens because the Cartoon and the Sketch domains can be expressed as approximate subsets of Photo and Art. So, discarding information that is specific to Photo and Art helps the classifier to be robust when applied to Cartoon or Sketch. However, this backfires when the evaluation is performed on Photo and Art, as the invariant representation is unable to capture the semantic diversity that their domain-specific information introduces. This effect can also be seen to be amplified in IB-IRM, which imposes a stronger commutative invariance across domains via information bottleneck.

**Orthogonality to flat-minima based algorithms:** Methods like Fishr (Rame et al., 2022; Rangwani et al., 2022), and Model Soups (Wortsman et al., 2022), all aim to achieve generalization across do-

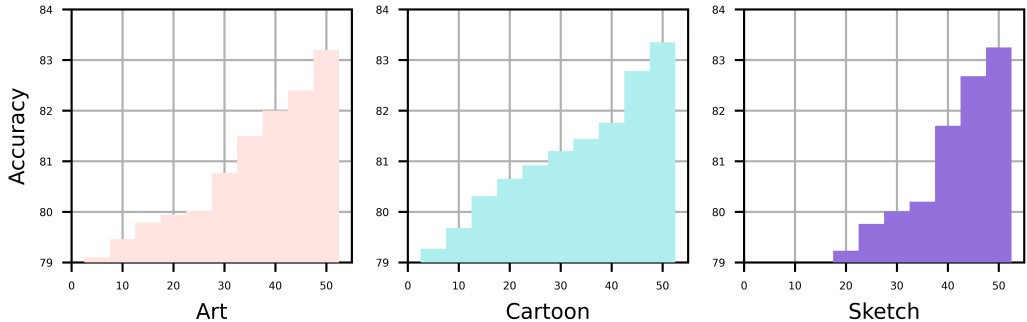

Figure 3: Effects of increasing the number of semantically complementary source samples (horizontal axis) on classification accuracy (vertical axis) across different complementary source domains.

mains by smoothing the loss landscape. For such algorithms, the domain asymmetry property, that the commutative invariance learning algorithms do not utilize as described above, do not seem to have a clear effect as illustrated in Table 2. Their performance is generally close to, or in some cases, better than the best of both classes of algorithms – ones that do (eg. DANN) and the ones that do not (eg. ERM) learn any form of invariance. Our conjecture is that loss landscape smoothing has the effect of mapping all the domains to a unified landscape with a prominent global minimum, which aids performance across all domains. When combined with NCI, the gains obtained on top of such algorithms do not seem to depend upon the degree of shared / domain-specific information, suggesting that the idea of commutativity is orthogonal to the flatness of the loss landscape. Specifically, if one were to choose domain-specific / shared information as the criterion, the gains provided by these methods seem to be random. This, however, is in addition to NCI providing non-trivial accuracy boosts when combined with the loss landscape smoothing algorithms, which shows that these algorithms can also benefit from the idea of non-commutativity. The a priori reasons behind these observations, unfortunately, fall outside of the scope of this work; something we would like to leave as an open problem at the intersection of invariance learning and function smoothness.

**Varying degrees of complementarity across domains:** Theorem 2 shows that NCI is particularly helpful when the semantic information provided by different domains is complementary to each other, enabling source domain samples to act as augmentations for meeting the sample complexity needs of learning the optimal target domain encoder. To evaluate this empirically, we conduct experiments on the PACS dataset with "Photo" as the target domain, where we start by sharing 50% of data support (sample IDs) across all domains. We designate one of the source domains as the "complementary source". Retaining the originally shared 50%, we progressively incorporate additional samples from the complementary source, varying from 5-50%, while removing the same amount of shared samples from the complementary source train set. This keeps the total number of instances in the train set constant, while increasing the degree of complementarity between the target and the complementary source. We repeat this experiment by assigning different domains as the complementary source, and report our results in Figure 3, where the horizontal axis represents the varying degrees of complementarity from 5-50%, and the vertical axis shows classification accuracy.

We observe that as we increase the degree of complementarity, the classification accuracy approaches the setting where the model has access to all the corresponding samples from the target (as well as the other source domains) (Table 1). The rise in the classification accuracy is smoother for domains that are similar to photos, *i.e.*, Art and Cartoon (perhaps because both have color information), and is more abrupt for sketches, which is relatively more disjoint from photos. However, once the total number of samples cover the full support (50% complementarity), the accuracies across all the complementary sources are similar and close to the maximum attainable value in Table 1. This shows that NCI can leverage semantically complementary samples from any of the source domains, and make them act as augmentations in the representation space of the target, achieving similar performance as that of using all target-specific samples.

**Discovering semantic asymmetries:** One of the conditions under which NCI has an advantage is that of the existence of semantic asymmetries – semantically relevant factors of variation that are present in some domain, but not in the others, as we formalize in Assumption 1. We test the existence of these factors by performing an additional set of experiments on the multi-modal material

| Backbone | Standalone | | | | NCI | | | | Oracle |
|---|---|---|---|---|---|---|---|---|---|
| | **Image** | **AOLP** | **DOLP** | **NIR** | **Image** | **AOLP** | **DOLP** | **NIR** | |
| TopFormer (CVPR'22) | 45.10 | 33.60 | 29.80 | 28.77 | **48.10** | 39.30 | 35.82 | 29.60 | 48.44 |
| DPT (ICCV'21) | 37.86 | 30.55 | 23.86 | 23.40 | **42.72** | 32.33 | 27.52 | 25.23 | 43.50 |
| SegFormer (NeurIPS'21) | 41.81 | 24.90 | 29.00 | 22.99 | **46.37** | 34.33 | 36.24 | 35.81 | 47.25 |
| FPN (CVPR'19) | 30.20 | 28.68 | 33.21 | 26.79 | 39.86 | 32.79 | **41.35** | 29.22 | 41.68 |
| DeepLabV3+ (ECCV'18) | 35.86 | 27.76 | 22.56 | 25.87 | **41.96** | 30.76 | 29.82 | 30.50 | 42.90 |
| PSPNet (CVPR'17) | 27.99 | 25.71 | 32.50 | 23.60 | 37.68 | 30.50 | **39.20** | 27.45 | 40.06 |
| UNet (MICCAI'15) | 29.00 | 26.02 | 23.51 | 21.27 | **36.93** | 31.58 | 27.29 | 25.63 | 37.70 |

Table 3: Left: Discovering semantic asymmetries to inform the direction of invariance learning in NCI. Right: Comparison with oracle performance.

segmentation dataset, the results of which we report in Table 3. We hypothesise that if the distribution of semantic information is asymmetric across domains, then the standalone performance of single domain training should vary across domains. This domain where the model performs the best should be the one that contains the most domain-specific information that is semantically relevant. Under circumstances where the cost of acquiring data from each domain is the same, this insight can be used to determine the most optimal direction for learning non-commutative invariances. The target domain for NCI in that case should be the domain that provides the highest accuracy with standalone training. From Table 3, it can be seen that for a given model, the performance of NCI correlates with standalone performance. The domain that gives the highest standalone accuracy (underlined) is also the one that has the highest accuracy when it becomes the target domain for NCI. This observation holds for all backbones and domains, indicating the validity of our hypothesis.

**Comparison with oracle performance:** To empirically validate whether NCI is, in fact, the correct form of invariance to learn, we compare its performance with that of an oracle, across a number of backbones. We choose the task of multimodal material segmentation, which is a recently emerging image segmentation task that requires leveraging material information from multiple imaging modalities to segment material types (Liang et al., 2022; Zhang et al., 2023a). A multi-modal material segmentation model can be viewed as an oracle, since it has access to all the domains (modalities for a multi-modal model) at test-time, allowing it to choose from meaningful and superfluous features that may or may not be specific to the domain. The ability of the model to make predictions based on multiple input modalities thus provides a unique setup to simulate the oracle behavior. Note that our setup of evaluating domain invariant features do not assume access to multiple input modalities at test time. The ordinary (non-oracle) models of NCI only have access to the target domain (single modality) data at test time. On the other hand, the oracle model has the advantage of accessing information across all modalities, both source and target during test. Specifically, we choose the model by Liang et al. (2022) to maintain consistency in terms of the network backbone. From Table 3, the performance of NCI can be seen to approach that of the oracle with access to just a single modality, indicating that it is able to capture the semantically meaningful invariances.

## 5 CONCLUSION

We presented the idea of learning non-commutative invariances (NCI), an invariance learning approach that is conditioned upon the target domain of inference. Along with the shared concepts across domains, NCI preserves domain-specific information for the target only. We prove this approach to be theoretically beneficial for domain adaptation when - (1) the target domain contains semantically relevant information that is not present in the source domain, a condition that we call Asymmetry (Assumption 1); and (2) the source domain samples are semantically complementary to those from the target domain, which allows NCI to satisfy the sample complexity needs for the optimal target domain encoder, with additional samples from the source domain (Theorem 2). Under these conditions, we empirically showed that NCI achieves SOTA performance in domain adaptation relative to existing invariance learning objectives. NCI is a novel conditional invariance learning paradigm, and as such, leaves room for a number of open questions: (1) Is the NCI solution of learning $\Phi_\tau^*$ unique? (2) Does $\varphi_\tau^* \to \Phi_\tau^*$ in a measure theoretic sense, and not just in terms of sample complexity or error bound? (3) What is the relationship between class-conditional invariance learning and NCI? (4) Is NCI provably orthogonal to flat-minima based algorithms? We believe that these can be explored as both theoretical and empirical research problems on invariant representations.

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

## A  APPENDIX

### A.1  EXTENDED LITERATURE REVIEW

**Augmentations with Domain Shift:** A number of works have leveraged domain adaptation to use easy-to-acquire source domain samples as data augmentations while training with target domain data. This is one of the primary modes of data-augmentation in the task of robotic grasping (Tobin et al., 2017; Bousmalis et al., 2018; Tanwani, 2021), where simulated data are used as an augmentation source along with real data, which is the target domain. This idea of leveraging simulated samples as source-side augmentations has also found application in a number of vision problems such as classification (Gan et al., 2021; Mishra et al., 2022), object detection (Peng et al., 2015; Tobin et al., 2017; Prakash et al., 2019; Tanwani, 2021), segmentation (Ros et al., 2016; Wang et al., 2020; Hoyer et al., 2023), and depth estimation (Zheng et al., 2018; Chen et al., 2019; Akada et al., 2022). Most training objectives for such applications can be classified into the self-supervised (Li et al., 2019; Hoyer et al., 2023) or the adversarial invariance learning (Tsai et al., 2018; 2019) categories, some of which can provide gains that are complementary to existing task-specific algorithms (Tsai et al., 2019). Via Theorem 2, we show that our proposed NCI can provably leverage source domain samples as augmentations in the target domain, and hence, can possibly benefit all of these application areas.

### A.2  NOTATIONS

- $\mathcal{D}_s$: Source domain, only available during training.
- $\mathcal{D}_\tau$: Target domain in which the model is to be evaluated.
- $X, Y, P(X, Y)$: Inputs, outputs, and their probability distribution respectively.
- $\mathcal{H}_\eta$: Hypothesis class (set of learnable functions) of domain discriminators.
- $\mathcal{H}_\varphi$: Hypothesis class of encoders for domain adaptation.
- $\eta$: Domain discriminator that distinguishes between source and target domains.
- $\Phi_\tau$: Encoder trained and evaluated in the target domain.
- $\Phi_s$: Encoder trained and evaluated in the source domain.
- $\Phi_\tau^*$: Optimal encoder for the target domain (trained only with target domain data).
- $\varphi$: Encoder trained with data from multiple domains.
- $\varphi^*$: Encoder that is optimal-on-average across domains (trained with multi-domain data).
- $\varphi_\tau$: Encoder trained with multi-domain data, but evaluated on domain $\mathcal{D}_\tau$.
- $\varphi_\tau^*$: Optimal encoder for $\mathcal{D}_\tau$, trained with multi-domain data.

Note that we have separate symbols for $\Phi_\tau^*$ and $\varphi_\tau^*$ only to distinguish that the former only sees samples from the target domain during training, while the latter can be trained on samples from multiple domains (including the target). Under certain circumstances like NCI, $\varphi_\tau^* \equiv \Phi_\tau^*$.

### A.3  ADDITIONAL PRELIMINARIES

#### A.3.1  METRIC SPACE OF DOMAINS

The set of domains $\mathbb{D}$ form a metric space under a hypothesis class of domain discriminators $\mathcal{H}_\eta$. The associated distance metric between any two points (domains) $s, \tau \in \mathbb{D}$ is their corresponding $\mathcal{H}_\eta$-divergence (Definition 5). It can be verified from Equation (4) that under any arbitrary encoder $\varphi$ that preserves the semantics of the input, $\mathcal{H}_\eta$-divergence satisfies the axioms for being a metric in $\mathbb{D}$, *i.e.*, for all domains $s, m, \tau \in \mathbb{D}$:

- Positivity: $\forall x \neq y, \mathcal{H}_\eta(s, \tau, \varphi) > 0$
- Symmetry: $\mathcal{H}_\eta(s, \tau, \varphi) = \mathcal{H}_\eta(\tau, s, \varphi)$
- Triangle inequality: $\mathcal{H}_\eta(s, \tau, \varphi) = \mathcal{H}_\eta(s, m, \varphi) + \mathcal{H}_\eta(m, \tau, \varphi)$

Hence, the distance $\Delta$ between any pair of source ($s$) and target ($\tau$) domains in the representation space of an encoder $\varphi$ can be measured via their $\mathcal{H}_\eta$-divergence as:

$$\Delta = s - \tau = \mathcal{H}_\eta(s, \tau, \varphi)$$

### A.3.2 OPERATOR-ENCODER DUALITY

We introduce the notion of operators to summarize a learning objective in the form of an expression in terms of domains. This allows us to make statements about (e.g., Theorem 4) and invoke (e.g., Theorem 2) group-theoretic properties that are induced by an invariance learning algorithm on the set of domains. In our analysis, an operator corresponds to a function that can be applied to one or more domains for performing some downstream task. In practical implementation, the operator would simply correspond to the data encoder network $\varphi$ that produces feature representations given an input $\mathbf{x}$ as $\varphi(\mathbf{x})$. The operator expression corresponding to the encoder summarizes the behaviour of that encoder when applied to data from different domains, which therefore establishes a duality between an encoder and its corresponding operator. In other words, the kind of function that is learned by the algorithm can be expressed via its operator expression. For instance, the learning objective of Domain Adversarial Neural Networks or DANNs (Ganin et al., 2016) can be written as a duality between a commutatively invariant operator $\otimes$ operating on samples from multiple domains (say, $\mathcal{D}_s$ and $\mathcal{D}_\tau$), and an optimal encoder $\varphi^*$ for the shared concept $\mathbf{x}$ between the domains as:

$$\mathbf{x}_s \otimes \mathbf{x}_\tau = \mathbf{x}_\tau \otimes \mathbf{x}_s = \varphi^*(\mathbf{x})$$

The above equality means that the function learned by a DANN is invariant to both the source and the target domain, producing the same output that corresponds to a domain-invariant encoding of the underlying concept $\mathbf{x}$. The fact that DANNs treat both the source and the target domains as the same is reflected through the commutative nature of $\otimes$ – the same output is produced irrespective of the position of $\mathbf{x}_s$ and $\mathbf{x}_\tau$ around $\otimes$. It also indicates (via $\varphi^*$) that a model trained with DANN is optimal on the shared concept $\mathbf{x}$ across the input domains in the operand list ($\mathbf{x}_s$ and $\mathbf{x}_\tau$) of the expression. On the other hand, the operator expression for NCI could be written in terms of a non-commutatively right invariant operator as:

$$\varphi_s^*(\mathbf{x}_s) = \mathbf{x}_s \otimes \mathbf{x}_\tau \neq \mathbf{x}_\tau \otimes \mathbf{x}_s = \varphi_\tau^*(\mathbf{x}_\tau)$$

Note that unlike DANN, $\otimes$ represents a non-commutatively right invariant operator for NCI. The above expression indicates that an encoder $\varphi_{s/\tau}^*$ trained with NCI is invariant to the domain that appears to the right of $\otimes$, but is sensitive to the domain that appears to the left. It also says that the encoder that NCI produces is optimal in the domain that the left operand comes from.

In summary, the operator expression summarizes the invariance properties of a learning algorithm, whereas the encoder is the actual function (say, a neural network) that computes such invariants, given the input data.

### A.3.3 RISKS

In general, our theoretical analysis (for instance, in Lemma 3, Theorem 1, and Theorem 3) deals with the empirical risk (Vapnik, 1991) defined as:

$$R(f) = \frac{1}{N} \sum_{i=1}^{N} l\left(f\left(\mathbf{x}_i\right), \mathbf{y}_i\right),$$

where $f$ could be an encoder $\varphi$ or a discriminator $\eta$, and $N$ is the number of train set samples in $\mathcal{D}_s \cup \mathcal{D}_\tau$. The complete expansion of $l(\cdot, \cdot)$ depends on the nature of $f$. As described in Section 3.4, $l(\cdot, \cdot)$ takes the form of Equation (3) for encoders $\varphi$, and that of Equation (2) for discriminators $\eta$.

### A.3.4 DEFINITIONS

The generalization of a model across multiple domains is governed by the No Free Lunch Theorem (Wolpert & Macready, 1997). As such, it is well known that models that perform well on average are not optimal for specific domains, and vice versa (Eastwood et al., 2022). Inspired by this, we present the following domain-specific and domain-agnostic notions of optimality.

**Definition 3** (Domain-Specific Optimality). An encoder $\Phi^*$ is said to be optimal for the domain $\mathcal{D}_\tau$ *iff*, for all $\Phi \in \mathcal{H}_\Phi$ (hypothesis class of domain-specific encoders) operating on $\mathcal{D}_\tau$, the following holds:

$$R(\Phi^*) = \int\limits_{X,Y} l(\Phi^*(\mathbf{x}), \mathbf{y})\, dP(\mathbf{x}, \mathbf{y}) \leq \int\limits_{X,Y} l(\Phi(\mathbf{x}), \mathbf{y})\, dP(\mathbf{x}, \mathbf{y}),$$

where the samples $\mathbf{x} \sim X$, $\mathbf{y} \in Y$ are all drawn from the domain $\tau$. In other words, there exits no other encoder $\Phi \in \mathcal{H}_\Phi$ with a true risk lower than what is achieved by $\Phi^*$ when operating in $\mathcal{D}_\tau$.

**Definition 4** (Optimal-on-average). An encoder $\varphi^*$ is said to be optimal on average across domains $[\tau, \tau + \Delta]$ *iff*, for all $\varphi \in \mathcal{H}_\varphi$, the following holds:

$$R(\varphi^*) = \frac{1}{\Delta} \int\limits_{\theta=\tau}^{\tau+\Delta} l(\varphi^*(\mathcal{D}_\theta), Y)\, dP(\theta, Y) \leq \frac{1}{\Delta} \int\limits_{\theta=\tau}^{\tau+\Delta} l(\varphi(\mathcal{D}_\theta), Y)\, dP(\theta, Y),$$

where we use $\varphi(\mathcal{D}_\theta)$ to denote the application of $\varphi$ to all elements of $\mathcal{D}_\theta$ for notational simplicity. In other words, all encoders $\varphi \in \mathcal{H}_\varphi$ should have a true risk that is at least as high as $R(\varphi^*)$ when averaged across $[\tau, \tau + \Delta]$.

**Definition 5** ($\mathcal{H}_\eta$-divergence (Ben-David et al., 2006; 2010; Kifer et al., 2004; Ganin et al., 2016)). The $\mathcal{H}_\eta$-divergence between $\mathcal{D}_s$ and $\mathcal{D}_\tau$ is a measure of the capacity of the discriminator hypothesis class $\mathcal{H}_\eta$ to distinguish between the source and the target sample representations. It is quantified as:

$$d_{\mathcal{H}_\eta}(\mathcal{D}_s, \mathcal{D}_\tau, \varphi) = 2 \sup_{\eta \in \mathcal{H}_\eta} \left| \Pr_{\mathbf{x} \sim \mathcal{D}_s} [\eta(\varphi(\mathbf{x})) = 1] - \Pr_{\mathbf{x} \sim \mathcal{D}_\tau} [\eta(\varphi(\mathbf{x})) = 1] \right| \tag{4}$$

In other words, the $\mathcal{H}$-divergence is a measure of the capacity of $\mathcal{H}$ to distinguish between samples in $\varphi(\mathcal{D}_s)$ from the samples in $\varphi(\mathcal{D}_\tau)$. An empirical estimate of the $\mathcal{H}_\eta$-divergence was also developed by Ben-David et al. (2006; 2010), which is defined when $\mathcal{H}_\eta$ is symmetric [1], with samples $S \sim (\mathcal{D}_s)^n$ and $T \sim (\mathcal{D}_\tau)^n$, as:

$$\hat{d}_{\mathcal{H}_\eta}(S, T, \varphi) = 2 \left( 1 - \min_{\eta \in \mathcal{H}_\eta} \left[ \frac{1}{n} \sum_{i=1}^n I[\eta(\varphi(\mathbf{x}_i)) = 0] + \frac{1}{n'} \sum_{i=n+1}^N I[\eta(\varphi(\mathbf{x}_i)) = 1] \right] \right) \tag{5}$$

where $I[a]$ is the indicator function that takes a value of 1 when the predicate $a$ is true, and 0 otherwise, and $S = \{\mathbf{x}_1, \mathbf{x}_2, ..., \mathbf{x}_n\}$, $T = \{\mathbf{x}_{n+1}, \mathbf{x}_{n+2}, ..., \mathbf{x}_N\}$.

**Definition 6** ($\mathcal{A}$-distance Ben-David et al. (2006); Ganin et al. (2016)). The $\mathcal{A}$-distance between domains $\mathcal{D}_S^X$ and $\mathcal{D}_T^X$ is an empirical approximation of the $\mathcal{H}$-divergence given by:

$$\hat{d}_{\mathcal{A}}(\mathcal{D}_S^X, \mathcal{D}_T^X) = 2 \sup_{A \in \mathcal{A}} \left| \Pr_{\mathcal{D}_S^X}(A) - \Pr_{\mathcal{D}_T^X}(A) \right|$$

$$\approx 2(1 - \epsilon) \implies \epsilon = 1 - \frac{\hat{d}_{\mathcal{A}}(\mathcal{D}_S^X, \mathcal{D}_T^X)}{2}$$

where $\mathcal{A}$ is a subset of $X$, and $\epsilon$, which approximates the "min" part of Equation (5), can be viewed as the generalization error Ganin & Lempitsky (2015).

**Theorem 3** (Generalization Bound on the Target Risk (Ben-David et al., 2006)). *Let the VC (Vapnik–Chervonenkis) dimension (Vapnik & Chervonenkis, 1971) of $\mathcal{H}_\eta$ be d. With probability $1 - \delta$ over the choice of samples $S \sim \mathcal{D}_s$ and $T \sim \mathcal{D}_\tau$, for every $\eta \in \mathcal{H}_\eta$:*

$$R_{\mathcal{D}_\tau}(\varphi) \leq R_S(\varphi) + \sqrt{\frac{4}{n}\left(d \log \frac{2en}{d} + \log \frac{4}{\delta}\right)} + \hat{d}_{\mathcal{H}_\eta}(S, T, \varphi) + 4\sqrt{\frac{1}{n}\left(d \log \frac{2n}{d} + \log \frac{4}{\delta}\right)} + \beta \tag{6}$$

*with $\beta \geq \inf_{\varphi^* \in \mathcal{H}_\varphi}[R_{\mathcal{D}_s}(\varphi^*) + R_{\mathcal{D}_\tau}(\varphi^*)]$, $n = |S| = |T|$, and*

$$R_S(\varphi) = \frac{1}{n} \sum_{i=1}^n I[\varphi(\mathbf{x}_i) \neq y_i]$$

*is the empirical source risk.*

---

[1] A hypothesis class $\mathcal{H}$ is symmetric *iff* $\forall h \in \mathcal{H}$ and any permutation of labels $c : Y \to Y$, we have $c(h) \in \mathcal{H}$ (Ganin et al., 2016).

Ganin et al. (2016) minimized the RHS of the bound by minimizing $\hat{d}_{\mathcal{H}_\eta}$ via domain adversarial training (a form of commutative invariance learning), which achieves a lower bound of 0 on the generalization error $\epsilon$ (Definition 6). However, we show in the section below, that such a solution is not optimal, as there exists an $\eta$, for which, although the generalization error $\epsilon$ is not 0, the $\mathcal{H}_\eta$-divergence is 0, without affecting the source risk $R_s$.

## A.4 PROOFS

**Result 1.** If the VC (Vapnik–Chervonenkis) dimension (Vapnik & Chervonenkis, 1971) of $\mathcal{H}_\varphi$, $VC(\mathcal{H}_\varphi) \geq N$, the total number of training samples, then the target risk of the optimal NCI encoder $R_{\mathcal{D}_\tau}(\varphi_\tau^*)$ is related to the target risk of the commutatively invariant encoder $R_{\mathcal{D}_\tau}(\varphi^*)$ as:

$$R_{\mathcal{D}_\tau}(\varphi_\tau^*) \leq R_{\mathcal{D}_\tau}(\varphi^*) - \hat{d}_{\mathcal{H}_\eta}(S, T, \varphi^*) < R_{\mathcal{D}_\tau}(\varphi^*)$$

*Proof.* If the VC (Vapnik–Chervonenkis) dimension (Vapnik & Chervonenkis, 1971) of $\mathcal{H}_\varphi$, $VC(\mathcal{H}_\varphi) \geq N$, then for the optimal commutative adversarial hypothesis $\varphi^*(\cdot)$ acting against the optimal discriminator $\eta^*$ [2] obtained via the minimization criterion in Equation (5), we have:

$$\frac{1}{n} \sum_{i=1}^n I[\eta^*(\varphi^*(\mathbf{x}_i)) = 0] + \frac{1}{n'} \sum_{i=n+1}^N I[\eta^*(\varphi^*(\mathbf{x}_i)) = 1] = 0$$

In other words, $\varphi^*(\cdot)$ would produce representations such that all samples from the source domain would be misclassified as being from the target domain and vice versa. So, under commutativity samples from both domains are equally likely to get misclassified by the domain classifier, as for example in Ganin & Lempitsky (2015); Arjovsky et al. (2019); Ahuja et al. (2021). Hence, the best possible $\mathcal{H}$-divergence that can be obtained with $\varphi'(\cdot)$ by substituting the above in Equation (5) is:

$$\hat{d}_{\mathcal{H}_\eta}(S, T) = 2(1 - [0 + 0]) = 2$$

However, for the optimal encoder $\varphi_\tau^* \in \mathcal{H}_\varphi$ that is non-commutatively invariant towards the target domain $\tau$, the samples from the source would be classified as being from the target, but not the other way around. We would thus have:

$$\frac{1}{n} \sum_{i=1}^n I[\eta^*(\varphi_\tau^*(\mathbf{x}_i)) = 1] = 1 \quad \text{and} \quad \frac{1}{n'} \sum_{i=n+1}^N I[\eta^*(\varphi_\tau^*(\mathbf{x}_i)) = 1] = 0$$

Substituting the above in Equation (5):

$$\hat{d}_{\mathcal{H}_\eta}(S, T) = 2(1 - [1 + 0]) = 0$$

Hence, under non-commutativity, *i.e.*, when the adversary forces only the misclassification of the domain labels of samples from the source domain, the theoretical limit of the $\mathcal{H}_\eta$-divergence is lower than that of the commutative setting, and in fact degenerates to 0, giving a stricter upper bound on the target risk in Theorem 3. Under NCI, Equation (6) thus becomes:

$$R_{\mathcal{D}_\tau}(\varphi_\tau^*) \leq R_s(\varphi_\tau^*) + \sqrt{\frac{4}{n}\left(d \log \frac{2en}{d} + \log \frac{4}{\delta}\right)} + 4\sqrt{\frac{1}{n}\left(d \log \frac{2n}{d} + \log \frac{4}{\delta}\right)} + \beta$$

$$\implies R_{\mathcal{D}_\tau}(\varphi_\tau^*) \leq R_{\mathcal{D}_\tau}(\varphi^*) - \hat{d}_{\mathcal{H}_\eta}(S, T, \varphi^*) < R_{\mathcal{D}_\tau}(\varphi^*)$$

as $R_s(\varphi_\tau^*) = R_s(\varphi^*)$ (non-commutativity does not affect the source risk as source samples still get mapped to the target domain, as with commutativity), and $\hat{d}_{\mathcal{H}}(S, T) = 0$, thereby bringing down the upper-bound on the target risk of the optimal NCI encoder to $R(\varphi^*) - \hat{d}_{\mathcal{H}_\eta}(S, T, \varphi^*)$. This completes the proof. $\qquad \square$

**Lemma 1.** *Let $\mathcal{D}_s$ and $\mathcal{D}_\tau$ be separated by $d_{\mathcal{H}_\eta} > 0$. Then, the bases of $\mathcal{D}_s$ and $\mathcal{D}_\tau$ can be factorized as $\mathcal{D}_s = [\mathbb{C} \; \mathbb{D}_s]$ and $\mathcal{D}_\tau = [\mathbb{C} \; \mathbb{D}_\tau]$ such that:*

$$\mathbb{D}_s \cdot \mathbb{D}_\tau = 0$$

---

[2] An optimal discriminator $\eta^*$ maximizes Equation (5), while an optimal commutative adversarial encoder $\varphi^*$ minimizes Equation (5).

*Proof.* Although $\mathcal{D}_s$ and $\mathcal{D}_\tau$ are separated by $d_{\mathcal{H}_\eta}(\mathcal{D}_s, \mathcal{D}_\tau) > 0$, they share the same support, $(X, Y)$. Hence, they must have a subspace that corresponds to the shared concept, which is $\mathbb{C}$.

Now, let the complements of $\mathbb{C}$ in $\mathcal{D}_s$ be $\mathbb{D}_s$, and that in $\mathcal{D}_\tau$ be $\mathbb{D}_\tau$. Since a concept, and a domain-specific expression of the concept, can vary independently, $\mathbb{D}_s$ and $\mathbb{D}_\tau$ must be subspaces that are intrinsic to the domains, and hence, not shared between the domains (the extreme case is the scenario when the two domains share only the concept and no domain-specific components, for example, in texts and images (Federici et al., 2020)). Since the $\mathcal{H}_\eta$-divergence between $\mathcal{D}_s$ and $\mathcal{D}_\tau > 0$, it means that they are distinct domains. This implies that the distinctness of $\mathcal{D}_s$ and $\mathcal{D}_\tau$, which is expressed via their positive $\mathcal{H}_\eta$ divergence, must be a result of the fact that their respective domain-specific components, namely $\mathbb{D}_s$ and $\mathbb{D}_\tau$, do not share any bases. Hence, $\mathbb{D}_s$ and $\mathbb{D}_\tau$ must be independent $\implies \mathbb{D}_s \cdot \mathbb{D}_\tau = 0$. This completes the proof of the lemma. $\qquad\square$

**Lemma 2.** *For source and target domains $\mathcal{D}_s$ and $\mathcal{D}_\tau$ respectively with $d_{\mathcal{H}_\eta}(\mathcal{D}_s, \mathcal{D}_\tau) > 0$, the following holds for a commutatively invariant encoder $\varphi^*(\cdot)$:*

$$\varphi^*(\mathcal{D}_s \cdot \mathcal{D}_\tau^T) = \varphi^*(\mathcal{D}_s) \cdot \varphi^*(\mathcal{D}_\tau)^T$$

*Proof.* Let $\varphi^*(\mathbb{C}) = \hat{\mathbf{y}}$. Since $\varphi^*$ is commutatively invariant to all domains, let $\varphi^*(\mathbb{D}_1) = \varphi^*(\mathbb{D}_2) = \mathbf{k}$. Without loss of generality, let both $\hat{\mathbf{y}}$ and $\mathbf{k}$ be idempotent under self dot-product, *i.e.*, $\hat{\mathbf{y}}^n = \hat{\mathbf{y}}$ and $\mathbf{k}^n = \mathbf{k}$. Now, from Lemma 1, we can right the RHS in the lemma as,

$$\varphi^*(\mathcal{D}_s) \cdot \varphi^*(\mathcal{D}_\tau) = \varphi^*([\mathbb{C}\ \mathbb{D}_s]) \cdot \varphi^*([\mathbb{C}\ \mathbb{D}_\tau])^T = [\varphi^*(\mathbb{C})\ \varphi^*(\mathbb{D}_s)] \cdot [\varphi^*(\mathbb{C})\ \varphi^*(\mathbb{D}_\tau)]^T = [\hat{\mathbf{y}}\ \mathbf{k}]$$

and the LHS as,

$$\varphi^*(\mathcal{D}_s \cdot \mathcal{D}_\tau) = \varphi^*([\mathbb{C}\ \mathbb{D}_s][\mathbb{C}\ \mathbb{D}_\tau]^T) = \varphi^*([\mathbb{C}\mathbb{C}^T\ \mathbb{D}_s\mathbb{D}_\tau^T]) = [\varphi^*(\mathbb{C}\mathbb{C}^T)\ \varphi^*(\mathbb{D}_s\mathbb{D}_\tau^T)]$$

However, since $\mathbb{C}\mathbb{C}^T$ is in the same concept basis, namely that of $\mathbb{C}$, and should hence induce the same label prediction

$$\implies \varphi^*(\mathbb{C}) = \varphi^*(\mathbb{C}\mathbb{C}^T) = \hat{\mathbf{y}}$$

Now, $\mathbb{D}' = \mathbb{D}_s\mathbb{D}_\tau^T$ corresponds to the domains-specific components shared between $\mathcal{D}_s$ and $\mathcal{D}_\tau$. Given that $\varphi^*(\cdot)$ is invariant to any domain-specific information, if $\varphi^*(\mathbb{D}_s) = \varphi^*(\mathbb{D}_\tau) = k$,

$$\mathbb{D}' = \mathbb{D}_s\mathbb{D}_\tau^T \implies \varphi^*(\mathbb{D}_s\mathbb{D}_\tau^T) = k$$

We can therefore rewrite $\varphi^*(\mathcal{D}_s \cdot \mathcal{D}_\tau)$ as:

$$\varphi^*(\mathcal{D}_s \cdot \mathcal{D}_\tau) = [\varphi^*(\mathbb{C}\mathbb{C}^T)\ \varphi^*(\mathbb{D}_s\mathbb{D}_\tau^T)] = [\hat{\mathbf{y}}\ \mathbf{k}] = \varphi^*(\mathcal{D}_s) \cdot \varphi^*(\mathcal{D}_\tau)$$

This completes the proof of the lemma. $\qquad\square$

**Lemma 3.** *Let the $\mathcal{H}_\eta$-divergence between the source (s) and the target ($\tau$) domains be $\Delta$, i.e., $s = \tau + \Delta$. The risk of the optimal-on-average encoder $\varphi^*(\cdot)$ across domains is lower-bounded by its prediction error on the shared semantics between the source and the target domains given by:*

$$R(\varphi^*) = \min_{\varphi^* \in \mathcal{H}_\varphi} \frac{1}{\Delta} \int_{\theta=\tau}^{\tau+\Delta} l\left(\varphi^*(\mathcal{D}_\theta), Y\right) dP(\theta, Y) \geq \min_{\varphi^* \in \mathcal{H}_\varphi} l(\varphi^*(\mathcal{D}_s \cap \mathcal{D}_\tau), Y),$$

*where $l(\cdot, \cdot)$ is the loss function for measuring prediction error.*

*Proof.* Following from Lemma 1, for a given instance $\mathbf{x} \in X$, the shared semantics between domains must lie in the concept subspace, *i.e.*, $\mathcal{D}_s \cap \mathcal{D}_\tau \in \mathbb{C}$. Hence, the risk on the shared semantics can be written as:

$$l(\varphi^*(\mathcal{D}_s \cap \mathcal{D}_\tau), Y) = l(\varphi^*(\mathbb{C}), Y) \tag{7}$$

Let the integral in the lemma be denoted by $J$. Assuming the simplest case of domain discrepancy, where $\mathcal{D}_s$ and $\mathcal{D}_\tau$ are linearly related, considering the path on the domain manifold connecting the

two, $J$ can be lower-bounded by the risk on the dot-product between $\mathcal{D}_s$ and $\mathcal{D}_\tau$ under $\varphi^*(\cdot)$ which computes their similarity as follows:

$$J = \int_{\theta=\tau}^{\tau+\Delta} l\left(\varphi^*(\mathcal{D}_\theta), Y\right) dP(\theta, Y) \geq l(\varphi^*(\mathcal{D}_\tau) \cdot \varphi^*(\mathcal{D}_{\tau+\delta}), Y)$$

The linearization follows from the simplest form that the manifold can assume, and a dot-product in this regime would be an estimate of the similarity, *i.e.*, the degree of shared information between $\mathcal{D}_s$ and $\mathcal{D}_\tau$, which are nothing but domain-specific expressions of the support $(X, Y)$. Now, applying the factorization of $\mathcal{D}_1$ and $\mathcal{D}_2$ from Lemma 1 to the form of $J$ in Equation (8), we have:

$$J \geq l(\varphi^*([\mathbb{C}\ \mathbb{D}_1]) \cdot \varphi^*([\mathbb{C}\ \mathbb{D}_2]^T), Y) = l(\varphi^*([\mathbb{C} \cdot \mathbb{C}^T\ \mathbb{D}_1 \cdot \mathbb{D}_2^T]), Y) = l(\varphi^*([\mathbb{C}\mathbb{C}^T\ \mathbb{0}]), Y), \quad (8)$$

where the distributive property of $\varphi^*(\cdot)$ under the dot-product follows from Lemma 2. Combining Equation (7) and Equation (8), we get:

$$R(\varphi^*) = \min_{\varphi^* \in \mathcal{H}_\varphi} \frac{1}{\Delta} J \geq l(\varphi^*([\mathbb{C}\mathbb{C}^T\ \mathbb{0}]), Y) \equiv l(\varphi^*(\mathbb{C}), Y) = \min_{\varphi^* \in \mathcal{H}_\varphi} l(\varphi^*(\mathcal{D}_s \cap \mathcal{D}_\tau), Y),$$

since $\mathbb{C}$ and $\mathbb{C}\mathbb{C}^T$ represent the same concept subspace. This completes the proof of the lemma. $\quad\square$

**Theorem 1.** Let the $\mathcal{H}_\eta$-divergence between the source $(s)$ and the target $(\tau)$ domains be $\Delta$, *i.e.*, $s = \tau + \Delta$. Under asymmetry (Assumption 1), *i.e.*, $I(\varphi_s^*(\mathbf{x}_s); \mathbf{y}) < I(\varphi_\tau^*(\mathbf{x}_\tau); \mathbf{y})$, the risk of the optimal-on-average encoder $R(\varphi^*)$ for predicting $Y$ across all domains up to a distance $\Delta$ from the domain $\tau$ admits the following upper bound:

$$R(\varphi^*) = \min_{\varphi^* \in \mathcal{H}_\varphi} \frac{1}{\Delta} \int_{\theta=\tau}^{\tau+\Delta} l\left(\varphi^*(\mathcal{D}_\theta), Y\right) dP(\theta, Y) \geq l\left(\varphi_\tau^*(\mathcal{D}_\tau), Y\right),$$

where $l(\cdot, \cdot)$ calculates the encoding risk of a domain, and $\varphi_\tau^*(\cdot)$ is the optimal encoder for $\tau$.

*Proof.* From Lemma 3, we know:

$$R(\varphi^*) \geq \min_{\varphi^* \in \mathcal{H}_\varphi} l(\varphi^*(\mathcal{D}_s \cap \mathcal{D}_\tau), Y),$$

*i.e.*, the risk of the globally optimal encoder $R(\varphi^*)$ is upper-bounded by the information shared between the source and the target domains $l(\varphi^*(\mathcal{D}_s \cap \mathcal{D}_\tau), Y)$. Also, the global risk of $\varphi^*(\cdot)$ must be at least as large as the maximum risk in the individual domains, *i.e.*,

$$\min_{\varphi^* \in \mathcal{H}_\varphi} l(\varphi^*(\mathcal{D}_s \cap \mathcal{D}_\tau), Y) \geq \max[R_s(\varphi^*), R_\tau(\varphi^*)], \quad (9)$$

where $R_s(\cdot)$ and $R_\tau(\cdot)$ are risks evaluated in the source and target domains respectively. Since we are operating under asymmetry (Assumption 1) of $I(\varphi_s^*(\mathbf{x}_s); \mathbf{y}) < I(\varphi_\tau^*(\mathbf{x}_\tau); \mathbf{y})$, we also have:

$$I(\varphi_s^*(\mathbf{x}_s); \mathbf{y}) < I(\varphi_\tau^*(\mathbf{x}_\tau); \mathbf{y}) \implies l(\varphi^*(\mathcal{D}_s), Y) > l(\varphi^*(\mathcal{D}_\tau), \mathbf{y}) \implies R_s(\varphi^*) > R_\tau(\varphi^*)$$
$$(10)$$

Now, since $\varphi^*(\cdot)$ can do only as good as $I(\mathcal{D}_s, Y)$, and $I(\mathcal{D}_\tau, Y) > I(\mathcal{D}_s, Y)$ from the asymmetry assumption, it must be the case that:

$$\exists\, \varphi_\tau^* \in \mathcal{H}_\phi \mid l(\varphi^*(\mathcal{D}_\tau), Y) > l(\varphi_\tau^*(\mathcal{D}_\tau), Y) \implies R_\tau(\varphi^*) > R_\tau(\varphi_\tau^*) = l\left(\varphi_\tau^*(\mathcal{D}_\tau), Y\right) \quad (11)$$

Therefore, putting Lemma 3, Equation (9), Equation (10), and Equation (11) together, we get:

$$R(\varphi^*) \geq \min_{\varphi^* \in \mathcal{H}_\varphi} l(\varphi^*(\mathcal{D}_s \cap \mathcal{D}_\tau), y) \geq \max[R_s(\varphi^*), R_\tau(\varphi^*)]$$
$$= R_s(\varphi^*) > R_\tau(\varphi^*) > R_\tau(\varphi_\tau^*) = l\left(\varphi_\tau^*(\mathcal{D}_\tau), Y\right),$$

which completes the proof of the theorem. $\quad\square$

**Theorem 2.** Consider a function class $\mathcal{T}$ of transformations that defines target domains such that $\tau \sim \mathcal{T}, \mathbf{x}_\tau = \tau(\mathbf{x}), \mathbf{x}_\tau \sim \mathcal{D}_\tau$. Let $\Phi_\tau^*(\cdot)$ be the optimal encoder for the target domain with sample complexity for incurring an error of $\epsilon$ with probability $\delta$. From Haussler's theorem, the error incurred by the optimal target domain encoder $\Phi_\tau^*$ with probability $\delta$ and $M$ target domain samples is given by:

$$\epsilon \geq \frac{\ln |\mathcal{T}| + \ln \frac{1}{\delta}}{M},$$

With $m_\tau$ target domain samples and $m_s$ source domains samples, such that $M \leq m_s + m_\tau, M > m_\tau$, the optimal NCI encoder $\varphi_\tau^*(\cdot)$ achieves the same error bound $\epsilon$ with probability $\delta$.

*Proof.* Let $\{\cdot\}_n^s$ and $\{\cdot\}_n^\tau$ denote sets of $n$ source and target domain samples respectively. When using $\Phi_\tau^*$, given a fixed sample size $M$, the only variability in $\epsilon$ comes from the size of the function class $\mathcal{T}$. Now, $|\mathcal{T}|$ directly models the number of domain-specific parameters in $\mathcal{D}_\tau$. Hence, the train set samples must provide information specific to $\mathcal{D}_\tau$, for which purpose, they must be sampled from $\mathcal{D}_\tau$. Thus, to prove the theorem, one needs to effectively show that, under the given constraints on the number of available samples, with NCI, the source domain samples can be leveraged to bridge the sample complexity needs of the target domain.

Without loss of generality, consider the most conservative case where the number of training samples $M = m_s + m_\tau$. Then, the process of applying non-commutative invariance with $\varphi_\tau^*$ between the source $X_s$ and the target $X_\tau$ train sets directed towards the target domain $\tau$ can be expressed as follows:

$$X_s \otimes X_\tau = \{\mathbf{x}_s^1, \mathbf{x}_s^2, ..., \mathbf{x}_s^{m_s}\}_{m_s}^s \otimes \{\mathbf{x}_\tau^1, \mathbf{x}_\tau^2, ..., \mathbf{x}_\tau^{m_\tau}\}_{m_\tau}^\tau$$

Using Lemma 1, the sample vectors can be decomposed as:

$$X_s \otimes X_\tau = \{\mathbf{c}_s^1 \oplus \mathbf{d}_s^1, \mathbf{c}_s^2 \oplus \mathbf{d}_s^2, ..., \mathbf{c}^{m_s} \oplus \mathbf{d}_s^k\}_{m_s}^s \otimes \{\mathbf{c}_\tau^1 \oplus \mathbf{d}_\tau^1, \mathbf{c}_\tau^2 \oplus \mathbf{d}_\tau^2, ..., \mathbf{c}_\tau^m \oplus \mathbf{d}_\tau^m\}_{m_\tau}^\tau,$$

where $\mathbf{x} = \mathbf{c} \oplus \mathbf{d}$. The subscripts under the vectors $\mathbf{c}$ only represent the domains from which the samples containing these concepts are obtained. Since concepts are domain-agnostic, they do not contain any domain information. The concept and the domain-specific components can then be rearranged as follows:

$$X_s \otimes X_\tau = \left(\{\mathbf{c}_s^1, \mathbf{c}_s^2, ..., \mathbf{c}_s^{m_s}\} \oplus \{\mathbf{d}_s^1, \mathbf{d}_s^2, ..., \mathbf{d}_s^{m_s}\}^s\right)_{m_s}$$
$$\otimes \left(\{\mathbf{c}_\tau^1, \mathbf{c}_\tau^2, ..., \mathbf{c}_\tau^{m_\tau}\}_{m_\tau} \oplus \{\mathbf{d}_\tau^1, \mathbf{d}_\tau^2, ..., \mathbf{d}_\tau^{m_\tau}\}^\tau\right)_{m_\tau}$$

Now, since $\otimes$ is non-commutatively invariant to the target domain, the source domain-specific components can be removed from the above equation:

$$X_s \otimes X_\tau = \{\mathbf{c}_s^1, \mathbf{c}_s^2, ..., \mathbf{c}_s^{m_s}\} \otimes \left(\{\mathbf{c}_\tau^1, \mathbf{c}_\tau^2, ..., \mathbf{c}_\tau^{m_\tau}\}_{m_\tau} \oplus \{\mathbf{d}_\tau^1, \mathbf{d}_\tau^2, ..., \mathbf{d}_\tau^{m_\tau}\}^\tau\right)_{m_\tau}$$

Since $\oplus$ distributes over any invariance operator $\otimes$ [3], we have:

$$= \left(\{\mathbf{c}_s^1, \mathbf{c}_s^2, ..., \mathbf{c}_s^{m_s}\} \oplus \{\mathbf{d}_\tau^1, \mathbf{d}_\tau^2, ..., \mathbf{d}_\tau^{m_\tau}\}^\tau\right)_{m_s}$$
$$\otimes \left(\{\mathbf{c}_\tau^1, \mathbf{c}_\tau^2, ..., \mathbf{c}_\tau^{m_\tau}\}_{m_\tau} \oplus \{\mathbf{d}_\tau^1, \mathbf{d}_\tau^2, ..., \mathbf{d}_\tau^{m_\tau}\}^\tau\right)_{m_\tau}$$
$$= \{\mathbf{c}_s^1 \oplus \mathbf{d}_\tau^i, \mathbf{c}_s^2 \oplus \mathbf{d}_\tau^j, ..., \mathbf{c}^{m_s} \oplus \mathbf{d}_\tau^k\}_{m_s}^\tau \otimes \{\mathbf{c}_\tau^1 \oplus \mathbf{d}_\tau^1, \mathbf{c}_\tau^2 \oplus \mathbf{d}_\tau^2, ..., \mathbf{c}_\tau^m \oplus \mathbf{d}_\tau^{m_\tau}\}_{m_\tau}^\tau$$
$$= \{\mathbf{c}_s^1 \oplus \mathbf{d}_\tau^i, \mathbf{c}_s^2 \oplus \mathbf{d}_\tau^j, ..., \mathbf{c}^{m_s} \oplus \mathbf{d}_\tau^k, \mathbf{c}_\tau^1 \oplus \mathbf{d}_\tau^1, \mathbf{c}_\tau^2 \oplus \mathbf{d}_\tau^2, ..., \mathbf{c}_\tau^{m_\tau} \oplus \mathbf{d}_\tau^{m_\tau}\}_{m_s + m_\tau \geq M}^\tau,$$

such that $\{i, j, ..., k\} \in [1, m_\tau]$. In other words, the target domain components $\mathbf{d}_\tau$ that the source concept vectors $\mathbf{c}_s$ combine with, are the ones that are present in the target domain training samples. Therefore, from the above, the total number of samples in $|X_s \otimes X_\tau|$ is:

$$|X_s \otimes X_\tau| = m_s + m_\tau \geq M, \tag{12}$$

all of which belong to the target domain $\tau$. Now, according to the Haussler's theorem as mentioned in the statement of Theorem 2, the requirement of achieving an error bound of $\epsilon$ with probability $\delta$ is to have at least $M$ samples. This can be observed from the following rearrangement:

$$\epsilon \geq \frac{\ln |\mathcal{T}| + \ln \frac{1}{\delta}}{M} \implies M \geq \frac{\ln |\mathcal{T}| + \ln \frac{1}{\delta}}{\epsilon}$$

---

[3]Composition of $\oplus$ and $\otimes$ should either add or remove domain specific components.

Now from Equation (12), we know that non-commutatively combining $X_s$ with $X_\tau$ with $\otimes$ effectively results in $m_s + m_\tau \geq M$ samples in the target domain $\tau$. Therefore, $\otimes$ satisfies the sample complexity requirement of having access to at least $M$ samples for learning the optimal target domain encoder $\Phi_\tau^*$, that achieves an error bound of $\epsilon$ with probability $\delta$, when evaluated in the target domain $\tau$. This completes the proof of the theorem. □

**Theorem 4.** *If an operator $\otimes$ is commutatively invariant, it induces a commutative semigroup on the set of domains $\mathbb{D} = \{\mathcal{D}_1, \mathcal{D}_2, ..., \mathcal{D}_n\}$ that it is invariant to.*

*Proof.* Below, we show that $\mathbb{D}$ satisfies the axioms of a commutative semigroup under $\otimes$.

**Commutativity:** Since $\otimes$ is commutatively invariant, for any pair of domains $\mathcal{D}_i, \mathcal{D}_j$, by Definition 2 and Lemma 1, we have,
$$\mathcal{D}_i \otimes \mathcal{D}_j = \mathcal{D}_j \otimes \mathcal{D}_i = \mathcal{C},$$
where $\mathcal{C} = [\mathbb{C} \ \ \mathbb{0}] \in \mathbb{D}$ is the domain containing only the shared concepts across all $\mathcal{D} \in \mathbb{D}$ with no domain-specific information, and assuming $\mathbb{D}$ is closed under $\otimes$.

**Associativity:** For all domains $\mathcal{D}_i, \mathcal{D}_j, \mathcal{D}_k \in \mathbb{D}$, we have,
$$(\mathcal{D}_i \otimes \mathcal{D}_j) \otimes \mathcal{D}_k = \mathcal{C} \otimes \mathcal{D}_k = \mathcal{C}$$
$$\mathcal{D}_i \otimes (\mathcal{D}_j \otimes \mathcal{D}_k) = \mathcal{D}_i \otimes \mathcal{C} = \mathcal{C}$$
$$\therefore (\mathcal{D}_i \otimes \mathcal{D}_j) \otimes \mathcal{D}_k = \mathcal{D}_i \otimes (\mathcal{D}_j \otimes \mathcal{D}_k)$$

This completes the proof of the theorem. □

## A.5 EXPERIMENTAL DETAILS AND RESULTS

| | **Clipart** | **Inforgraph** | **Painting** | **QuickDraw** | **Real** | **Sketch** |
|---|---|---|---|---|---|---|
| ERM (Wiley 1998) | 44.24 | 43.68 | 46.79 | 39.31 | 37.86 | 40.50 |
| DANN (ICML'15) | 45.21 | 44.00 | 44.86 | 41.86 | 33.32 | 42.77 |
| CDANN (ECCV'18) | 45.35 | 44.82 | 45.10 | 41.50 | 35.37 | 42.36 |
| MDAN (NeurIPS'18) | 45.20 | 44.31 | 46.10 | 40.02 | 36.65 | 41.15 |
| MDD (ICML'19) | 44.68 | 44.21 | 45.68 | 41.02 | 35.91 | 40.86 |
| CICyc (GCPR'21) | 44.60 | 44.42 | 45.06 | 40.20 | 34.33 | 40.80 |
| IB-IRM (NeurIPS'21) | 45.55 | 44.68 | 43.21 | 41.95 | 32.86 | 42.88 |
| EQRM (NeurIPS'22) | 45.22 | 44.37 | 45.96 | 40.98 | 36.00 | 40.02 |
| CIRCE (ICLR'23) | 45.74 | 45.01 | 46.28 | 40.56 | 36.95 | 40.37 |
| SDAT+ELS (ICLR'23) | 45.68 | 44.88 | 46.45 | 41.28 | 36.37 | 42.09 |
| **NCI (Ours)** | **46.31** | **45.82** | **48.00** | **43.05** | **39.10** | **43.41** |

Table 4: Comparison with SOTA invariance learning approaches on DomainNet.

**Discussion on SOTA invariance learning algorithms:** Empirical Risk Minimization ERM Vapnik (1998) is the classical domain transfer approach which uniformly minimizes the empirical risk across all domains, still providing one of the best average performances. Ganin & Lempitsky (2015) introduced the idea of Domain Adversarial Neural Networks (DANN), to discover features that are shared between domains, which was extended in a number of later works including MDAN Zhao et al. (2018) and MDD Zhang et al. (2019). IB-IRM (Ahuja et al., 2021) explores the idea of information bottleneck for invariance learning, while EQRM (Eastwood et al., 2022) presented a probabilistic approach for generalizing across domains via quantile risk minimization. However, all of the above approaches learn some form of commutative invariance, wherein domain-specific information from both the source and the target are treated as being irrelevant. CDANN Li et al. (2018) was one of the earliest works that introduced the idea of conditional invariance learning into DANNs. Thereafter, CICyc (Samarin et al., 2021) enhanced this paradigm via disentanglement learning through cycle consistency. They proposed disentangling the latent representation into a property subspace $Z_0$ and an invariant subspace $Z_1$ encoding information about the label $Y$ and the domain respectively to achieve the conditional invariance of $X \perp\!\!\!\perp Y \mid Z_0$. This allows the generation of new $\tilde{X}$ of

a specific class by keeping $Z_0$ fixed and varying $Z_1$. However, such a disentanglement is based on the assumption that the property and the invariant (domain) subspaces are independent from each other. We show via Theorem 1 that such an assumption does not necessarily hold under domain asymmetry (Assumption 1). The empirical drawback of this assumption can be seen in Table 1 and Table 4. CICyc has strong performance for domains whose underlying transformations (Equation (1)) preserve asymmetrically high property information (e.g., Real, Photo, etc.), but relatively weaker performance on domains that aggressively diverge from other source domains, sharing only the concept at the abstract level (e.g., Sketch, QuickDraw). The idea of preserving domain specific information was also explored in Environment Label Smoothing (ELS) Zhang et al. (2023b), which is one of the most recent advancements in the domain adaptation literature to deal with environment label noise. ELS Zhang et al. (2023b) and CIRCE Pogodin et al. (2023) are two of the most recent advances in invariance learning for adapting to novel domains that achieve conditional invariance through environment label smoothing, and kernel-based conditional independence learning mechanisms respectively. However, unlike NCI, they require significant modifications of the invariance learning objectives, or inclusion of additional constraints that enforce conditional invariance, and do not necessarily guarantee optimality in the target domain.

