# OpenReview forum: "Learning Conditional Invariances through Non-Commutativity"
_ICLR.cc/2024/Conference — ICLR 2024 poster_

### Official Review · Reviewer_C1qz · 2023-10-25

**Soundness:** 2 fair
**Presentation:** 2 fair
**Contribution:** 3 good
**Rating:** 5
**Confidence:** 2

**Summary:**

The author present an idea of invariant learning, that is learning non-commutative invariances (NCI), which is conditioned on the target domain and therefore preserves target-specific information. When the target domain contains more semantically relevant information than the source domain, the authors theoretically show that the NCI approach is beneficial for domain adaptation. Empirical results are also provided and show the benefit of NCI approach.

**Strengths:**

The empirical results do show the benefit of the proposed NCI approach in domain adaptation tasks. The assumptions of asymmetry and non-commutative seems to be novel in the context of invariant learning. Also the authors aim to provide a general theoretical framework that captures the properties of invariant learning algorithms (through the lens of commutativity) and can explain the benefit of NCI.

**Weaknesses:**

The theory part of this work is very difficult to understand. Lack of necessary explanations of notations and definitions. See the questions for details.

**Questions:**

1. What is an optimal encoder $\Phi^{\star}$ (in what sense it is optimal, i.e., what is the objective function that it minimizes)?

2. In definition 1, what does an operator mean in the context of invariant learning (or domain adaptation)? Can you give an example of an operator in an invariant learning algorithm (e.g., DANN)?

3. You only give the definition of NCI for operators (in definition 1). What is a NCI encoder $\phi^{\star}$? Again, what does optimal mean in   Result 1?

4. In Theorem 1,  you write $s=\tau + \delta$. But how to do calculations between domain $s$ and $\tau$? You do not give any definition.

5. All the risks are not defined.

---

> ### Author Response · Authors · 2023-11-20
>
> We thank the reviewer for precisely pointing out the theoretical aspects of our work that require further clarification. To this end, we have introduced a number of subsections, primarily in the Appendix, that attempt to address such lack of clarity. Below, we discuss our changes in further detail.
>
> 0. **General theoretical clarity:** Through our update, we make the notions of optimality (**Appendix A.3.4, Definitions 3 and 4**) and risk (**Appendix A.3.3**) more explicit, elaborate upon the necessity of introducing the construct of operators and their connection with encoders (**Appendix A.3.2**), as well as formalize the notion of separation on the space of domains (**Appendix A.3.1**). Following the suggestions of Reviewer tufY, we also expand upon the proof of Theorem 2, disambiguate the usage of $\delta$ by introducing $\Delta$ as a separate symbol for inter-domain distance, as well as state the assumption of asymmetry within a separate mathematical environment.
>
> 1. **Optimality of** $\Phi^*$: We have now concretized all the notions of optimality, i.e., domain-specific ($\Phi^*$) and domain-agnostic ($\varphi^*$) in **Appendix A.3.4, Definitions 3 (Domain-Specific Optimality) and 4 (Optimal-on-Average)**, which details the exact conditions under which an encoder can be deemed as being optimal. From these definitions, it can be seen that the objective function that is minimized is the empirical risk, denoted by $R(\cdot)$, further details about which are provided in Section 3.4 (Training with NCI) of the main paper and Appendix A.3.3 (Risks).
>
> 2. and 3. **Meaning and example of operators, and their connection with encoders:** We have now added a dedicated subsection, **Appendix A.3.2 (Operator-Encoder Duality)**, that discusses all of these. It first establishes the motivation for introducing the notion of operators into our theoretical framework, and then discusses the connection between the abstract operators and more concrete / practical, real-world encoders (which are essentially neural networks in our case). It accompanies the discussion with examples, both from the paradigm of NCI (non-commutatively invariant operator) and DANN (commutatively invariant operator) to make the ideas more relatable to practice.
> ####
> 4. **Calculations in the space of domains:** We have now described in **Appendix A.3.1 (Metric Space of Domains)**, how the discrepancy between two domains can be calculated. In summary, the idea of $H_\eta$-divergence (Definition 5) allows us to quantify the separation between $s$ and $\tau$ as $s = \tau + \Delta$. In practice, this is calculated by the discriminator network $\eta$ that measures the separability of domains in the representation space of the encoder $\varphi$ through optimizing $\mathcal{L}_\eta$ (Equation 2).
>
> 5. **Risks:** We have now introduced **Appendix A.3.3 (Risks)** that explicitly defines the general form of the risks under consideration. In general, we minimize the empirical risk in all cases. Their specific forms depend on the kind of function that is being learned, as stated in Section 3.4 (Training with NCI) of the main paper.

---

> > ### Comment · Reviewer_C1qz · 2023-11-22
> > **Reply to authors**
> >
> > Thanks for your detailed explanation. I highly appreciate your attempt to address these issues in revised version, which resolves my concerns with respect to writing. Since I am not very familiar with this area, and may not fairly evaluate your contribution, I will raise my score to 5/6, with confidence 2. I hope other more knowledgable reviewers or AC can provide more accurate evaluations, and I will rely on their idea.

---

### Official Review · Reviewer_a8sS · 2023-10-29

**Soundness:** 3 good
**Presentation:** 3 good
**Contribution:** 3 good
**Rating:** 8
**Confidence:** 2

**Summary:**

This paper shows that a provably optimal and sampleefficient way of learning conditional invariances is by relaxing the invariance criterion to be non-commutatively directed towards the target domain. Both their theory and experiments show that non-commutative invariance can leverage source domain samples to meet the sample complexity of learning the optimal target-specific encoder, surpassing SOTA invariance learning algorithms.

**Strengths:**

1. The theoretical analysis is comprehensive.
2. The experimental results are solid.

**Weaknesses:**

none

**Questions:**

none

---

### Official Review · Reviewer_tufY · 2023-11-10

**Soundness:** 2 fair
**Presentation:** 3 good
**Contribution:** 3 good
**Rating:** 6
**Confidence:** 3

**Summary:**

This paper studies Non-Commutative Invariance (NCI) as an efficient way of learning conditional invariances. The authors propose a method that leverages domain-specific information and focuses on the target domain while learning. They argue that relaxing the invariance criterion to be non-commutatively directed towards the target domain results in a more optimal and sample-efficient learning of conditional invariances. The paper highlights the superiority of NCI over SOTA invariance learning algorithms for domain adaptation.

**Strengths:**

**S1** The paper is well-written, with math notations clearly explained.

**S2** The empirical observations are supported by theoretical guarantees.

**S3** The discussion surrounding each theorem makes the theoretical results more understandable.

**S4** The paper presents extensive numerical experiments and demonstrates that their NCI-based approach surpasses existing SOTA algorithms in invariance learning.

**Weaknesses:**

Please see below.

**Questions:**

Section 3.1:

I suggest presenting the following statement as a formal assumption within a mathematical environment for better clarity.

"For the remainder of our analysis, we assume that the target domain contains more semantically relevant information than the source domain, i.e., ..."

Section 3.2:

Please ensure consistency in the use of symbols across theorems. If the symbol $\delta$ in Theorems 1 and 2 refers to different variables, this should be corrected to avoid confusion.

Clarification in Proof: The proof of Theorem 2 requires more clarity, particularly in the concluding part i.e. “ … samples from the target domain, which is the … This completes the proof of the theorem”. It's currently difficult to understand how the final result is derived from the given explanation.

Related Work:

The technical comparison with the paper "Learning Conditional Invariance through Cycle Consistency" requires more detail.

---

> ### Author Response · Authors · 2023-11-20
> **Authors' Response to Reviewer tufY**
>
> We thank the reviewer for suggestions that not only improve the readability and comprehensibility of our paper, but also make our work more complete and thorough. To this end, we incorporate a number of changes throughout the main manuscript and the Appendix which we detail below.
>
> 1. **Section 3.1 Assumption:** We have now stated this as a separate, formal assumption as **Assumption 1 (page 3)**, with the corresponding mathematical formulation.
>
> 2. **Usage of $\delta$:** We thank the reviewer for pointing out this error. We have now corrected and replaced the notation for distance between domains with $\Delta$ across the full paper. So now, Theorem 1 and all associated results use $\Delta$ to denote distance between domains, whereas $\delta$ is used in Theorem 2 and all associated discussions to denote probability of success.
>
> 3. **Proof of Theorem 2:** We agree with the reviewer that the jump to the conclusion about the sample complexity of learning the optimal target domain encoder $\Phi^*_\tau$ was indeed too abrupt in the proof of Theorem 2. In **page 20**, we have now extended the proof of Theorem 2 to address this by explicitly specifying the link between how non-commutatively combining the source and the target domains maps all samples to the target domain, and the sample complexity of learning $\Phi^*_\tau$. If there is some aspect of the proof that is still unclear, we would be happy to have further inputs from the reviewer about the same.
>
> 4. **Technical comparison with Samarin et al., 2021:** We thank the reviewer for pointing this out, as despite discussing this work in the introduction as being very relevant to our paradigm of conditional invariance learning, we missed out on a more detailed technical comparison of our NCI with theirs. As part of this rebuttal, we have now addressed this by evaluating their method (dubbed CICyc in our manuscript) on the PACS, Office-Home, and DomainNet benchmarks, reporting the results in **Tables 1 and 4**. We have accompanied this with a technical discussion on their work and how our proposed theoretical framework can explain the empirical behaviour of their algorithm in **page 21**.

---

### Author Response · Authors · 2023-11-20

We thank the reviewers for their time in thoroughly reviewing our work, and for recognizing the novelty in our core proposition of non-commutativity for invariance learning. Through this rebuttal, we attempt to address all of their concerns, by incorporating their suggested corrections and clarifications into our paper. We have made a number of changes to our main paper and made significant additions to the Appendix section, highlighted with the color "blue", which we hope would add to the rigour of our work.

---

### Meta-Review · Area_Chair_6krP · 2023-12-12

**Metareview:**

The paper offers a novel methodological insight for domain adaptation, as well as improved experimental performance.

**Justification For Why Not Higher Score:**

The new insight is interesting but not groundbreaking.

**Justification For Why Not Lower Score:**

Methodological and empirical strength

---

### Decision · Program_Chairs · 2024-01-16

Accept (poster)